# Lower-Limb Muscle Power Is Negatively Associated with Protein Intake in Older Adults: A Cross-Sectional Study

**DOI:** 10.3390/ijerph192114579

**Published:** 2022-11-07

**Authors:** Hélio José Coelho-Júnior, Domenico Azzolino, Riccardo Calvani, Ivan de Oliveira Gonçalves, Matteo Tosato, Francesco Landi, Matteo Cesari, Anna Picca, Emanuele Marzetti

**Affiliations:** 1Department of Geriatrics and Orthopedics, Università Cattolica del Sacro Cuore, 00168 Rome, Italy; 2Department of Clinical and Community Sciences, University of Milan, 20122 Milan, Italy; 3Geriatric Unit, IRCCS Istituti Clinici Scientifici Maugeri, 20138 Milan, Italy; 4Fondazione Policlinico Universitario “Agostino Gemelli” IRCCS, 00168 Rome, Italy; 5Department of Health, Piaget University, Suzano 08673-010, Brazil; 6Department of Medicine and Surgery, LUM University, 70100 Casamassima, Italy

**Keywords:** nutrition, macronutrients, sarcopenia, frailty, physical function, elderly

## Abstract

The present study examined the association between lower-limb muscle power and protein-related parameters in older adults. This study followed a cross-sectional design. Participants were community-dwelling older adults. Candidates were considered eligible if they were 60 years or older, lived independently, and possessed sufficient physical and cognitive abilities to perform all the measurements required by the protocol. The 5 times sit-to-stand (5STS) test was performed as fast as possible according to a standard protocol. Absolute, relative, and allometric muscle power measures were estimated using 5STS-based equations. Diet was assessed by 24-h dietary recall and diet composition was estimated using a nutritional software. One-hundred and ninety-seven older adults participated to the present study. After adjustment for covariates, absolute and allometric muscle power were negatively associated with body weight-adjusted protein intake. Our findings indicate that absolute and allometric muscle power estimated through a simple equation are negatively associated with body weight-adjusted protein intake in community-dwelling older adults.

## 1. Introduction

Physical function refers to a large construct that involves numerous motor tasks that allow an individual to interact with the environment [1,2]. Physical function changes during the life course, such that it commonly increases during childhood, remains stable in adulthood, and decreases significantly past the fifth decade of life [1,3,4]. This scenario demands special attention because the preservation of physical function is a core element for maintaining independence and remaining engaged in social activities in advanced age [5,6]. Indeed, significant losses in physical function increase the risk of numerous negative outcomes, including cognitive decline, poor mobility, and death [7,8,9,10], have a central role in the development of frailty and sarcopenia [11,12], and negatively impact the prognosis of many diseases [13,14,15].

Muscle power is a type of physical function that encompasses the capacity of generating strength as quickly as possible [16,17]. In recent decades, studies have repeatedly indicated that muscle power declines earlier and faster with age than other important aspects of physical function (e.g., muscle strength) [18,19,20]. Furthermore, reduced muscle power is a strong, independent predictor of incident disability [20]. This evidence indicates that muscle power should be actively monitored during aging, and that strategies to maintain or even improve muscle power are urgently required [16,17].

The assessment of muscle power involves laboratory-based tests that are not completely adapted to the old population [21]. Available tests also lack standardized protocols, have high costs, and may possibly be harmful for older adults [21]. These limitations hamper the clinical assessment of muscle power in older adults. Recently, a simple equation was proposed by Alcazar et al. [22] to estimate lower-limb muscle power based on the time to complete the 5 times sit-to-stand (5STS) test, chair height, and the test person’s body weight (BW) and height. Muscle power estimates were validated against values obtained on a test performed in a leg press resistance machine [22]. Muscle power values estimated through the equation have been associated with quality of life, cognitive function, physical performance, and sarcopenia [22,23]. Although such associations were only tested cross-sectionally and need to be confirmed in longitudinal studies, 5STS-based muscle power estimation appears to be a feasible approach to assess lower-limb muscle power in clinical settings.

Nutrition is a modifiable lifestyle factor that may be harnessed to foster active and healthy aging [24,25]. In particular, protein intake greater than the current recommended dietary allowance (0.8 g/kg BW per day; RDA) may be proposed as a strategy to preserve physical function in advanced age [26,27,28,29]. More specifically, physically inactive older adults should be advised to ingest at least 0.8 g of protein/kg of BW/day, while a minimal intake of 1.2–1.5 g/kg of BW/day, depending on the type of physical exercise, is recommended to physically active people over 65 years [26]. These recommendations are supported by several studies showing that older adults with a high protein intake (HPI) have better performance on a set of physical performance tests, including the Short Physical Performance Battery (SPPB), greater lower-limb muscle strength, faster walking speed, and better balance than those who consume ≤0.8 g of protein/kg of BW/day [30,31].

The impact of protein intake on physical function is thought to be mediated, at least partly, by the anabolic action of branched-chain amino acids (BCAAs) on skeletal muscle [32]. In fact, adequate amounts of BCAAs, in particular leucine, are essential to optimally stimulate muscle protein synthesis (MPS) [32,33,34]. Recent evidence also suggests that the distribution of protein across the main meals might be more important than the total amount of protein ingested during the day [35,36,37].

To the best of our knowledge, only one study examined the association between protein intake and muscle power in older people [37]. The cross-sectional study involved 97 German healthy community-dwelling older adults without functional limitations and included measures of lower-limb muscle power, assessed using a pneumatic resistance seated leg press machine, and dietary intake, using 7-day food record. The authors reported that lower-limb muscle power was not significantly associated with mean daily protein intake, protein distribution across the meals, or number of meals providing at least 0.4 g of protein/kg of BW/per day [37].

To increase the knowledge on the subject, the present study examined the cross-sectional association between protein intake and measures of lower-limb muscle power (i.e., absolute, allometric, and relative) estimated using the equation proposed by Alcazar et al. [22] in community-dwelling Brazilian older adults. Our hypothesis was that muscle power and protein intake would be positively and significantly correlated.

## 2. Materials and Methods

This was a cross-sectional study that investigated the association between lower-limb muscle power estimates and protein-related parameters in community-dwelling Brazilian older adults. Lower-limb muscle power was estimated using 5STS-based equations [22] and food intake was assessed by 24-h dietary recall.

The study was approved by the Research Ethics Committee of the University of Mogi das Cruzes (UMC, São Paulo, Brazil). All study procedures were conducted in compliance with the Declaration of Helsinki and the Resolution 196/96 of the National Health Council. The manuscript was prepared in compliance with the STrengthening the Reporting of OBservational studies in Epidemiology (STROBE) guidelines for observational studies [38].

### 2.1. Participants

Participants were recruited between January 2015 and January 2018 in a community senior center located in the metropolitan area of São Paulo, Brazil. This study was advertised through posters placed in public sites (e.g., parks, city hall, public offices, bus stops, train stations), local radios, and newspapers. People were also invited to participate by direct contact. Candidates were considered eligible if they were 60+ year-old, lived independently, and possessed sufficient physical and cognitive abilities to perform all measurements required by the protocol. All participants provided written informed consent prior to enrolment.

### 2.2. Anthropometric Measures

An analog weight scale with a stadiometer (Filizola, Brazil) was used to measure BW and height. The body mass index (BMI) was calculated as the ratio between BW (kg) and the square of height (m^2^). Participants were classified as underweight, normal weight, or overweight according to the following cut-points: <22 kg/m^2^, 22–27 kg/m^2^, and >27 kg/m^2^ [39].

### 2.3. The 5 Times Sit-to-Stand Test

Participants were instructed to rise from a chair five times as quickly as possible with their arms folded across their chest. Timing began when participants raised their buttocks off the chair and was stopped when they were seated at the end of the fifth stand [1]. Absolute, relative, and allometric muscle power were estimated according to the equations proposed by Alcazar et al. [22]:(1)Absolute muscle power: ([Body weight × 0.9 × *g* × (height × 0.5 − chair height)])/(5STS (s) × 0.1);(2)Allometric muscle power: Absolute muscle power/height^2^;(3)Relative muscle power: Allometric muscle power/BMI.

### 2.4. Dietary Assessment

Food intake was assessed by 24-h dietary recall [40]. The method uses an open-ended questionnaire to provide a quantitative and subjective estimation of actual food consumption. In the present study, two trained researchers (H.J.C.-J. and I.O.G.) asked the participants to recall in detail all foods they consumed on a meal-by-meal basis, including snacks, during the previous 24 h. Interviews occurred on Tuesdays, Wednesdays, Thursdays, and Fridays to avoid possible biases associated with the weekend. Participants were requested to provide details about cooking methods (e.g., fried, grilled, and roasted), serving and portion sizes, product brands, sauces, spices, and condiments consumed, and the eventual use of dietary supplements. Amounts of beverages consumed were also recorded, and participants were asked to specify if and how beverages were sweetened. Two-dimensional aids (e.g., photos), household utensils (e.g., standard measuring cups and spoons), and food models were used to facilitate assessment of portion sizes. Diet composition was estimated using the NutWin software, version 1.5 (Federal University of São Paulo, Brazil) [41].

### 2.5. Statistical Analysis

Continuous variables are expressed as the mean ± standard deviation (SD) or absolute numbers (percentage). Pearson’s correlation analysis was conducted to investigate the association between muscle power- and protein-related parameters. Associations with a *p*-value lower than 0.05 were included in the regression analyses. The final model was adjusted for sex, age, BMI (continuous and categorical), and total kilocalories consumed per day. Significance was set at 5% (*p* value < 0.05) for all tests. All analyses were performed using the SPSS software (version 23.0, SPSS Inc., Chicago, IL, USA).

## 3. Results

### 3.1. Study Participants

Two-hundred and fifty-four candidates agreed to be evaluated for inclusion. Of these, 46 were younger than 60 years, nine had missing data for the 5STS test, and two had missing data for diet, leaving a total of 197 participants for the analysis. The main characteristics of study participants are shown in Table 1. The study population was mostly composed of female (83%) “young-older” adults (mean age: 68.3 years, range: 60 to 99 years). Most participants were overweight, regardless of sex (Appendix A) [39]. Mean 5STS performance scores were borderline to cutoff values for dynapenia [42]. Mean BW-adjusted protein intake was higher than the RDA (mean protein consumption = 1.5 g of protein /kg/BW per day). The majority of dietary protein was ingested at lunch.

### 3.2. Pearson’s Correlation Analysis

Results of Pearson’s correlation analysis are shown in Table 2. Absolute and allometric muscle power values were negatively and significantly associated with BW-adjusted protein intake and protein consumption at lunch. Relative muscle power was not significantly associated with either BW-adjusted protein intake or protein consumption at lunch. No significant associations were observed between any muscle power parameter and energy intake, absolute protein intake, protein intake at breakfast, lunch and dinner, or valine, isoleucine, and leucine intake.

### 3.3. Linear Regression Analysis

Results of the linear regression analysis are shown in Table 3. Unadjusted analyses indicated that BW-adjusted protein and protein intake at lunch were significantly associated with absolute muscle power. After adjusting the analysis according to age, sex, BMI (continuous), and mean daily energy intake, only BW-adjusted protein intake remained significantly associated with absolute muscle power. No protein-related parameter was significantly associated with allometric muscle power. When the analysis was run using BMI as a categorical variable (Appendix A), similar results were observed, given that BW-adjusted protein intake was significantly associated with absolute muscle power. However, BW-adjusted protein intake became significantly associated with allometric muscle power.

## 4. Discussion

The main findings of the present study indicate that lower-limb absolute muscle power was negatively associated with BW-adjusted protein in older adults, regardless of whether BMI was treated as a continuous or categorical covariable. No significant associations were observed between protein intake at lunch and absolute muscle power, or between protein-related parameters and allometric muscle power. A significant association was found between BW-adjusted protein and allometric muscle power when BMI was analyzed as a categorical variable.

Only a few studies have investigated the association between protein intake and muscle power in older adults; and, to the best of our knowledge, ours is the first investigation that examined this relationship using the recently validated equation of Alcazar et al. [22]. In contrast to our findings, Gingrich et al. [37] did not observe significant associations between BW-adjusted protein intake and lower-limb muscle power, assessed using a pneumatic resistance seated leg press machine, in German healthy community-dwelling older adults.

Our findings are quite unexpected. Adequate protein consumption is a major regulator of muscle metabolism, avoiding, or at least reducing muscle catabolism, by providing essential amino acids (AAs) equivalent to body demands [27,29,43]. AAs stimulate MPS through the activation of the mammalian target of rapamycin and its downstream proteins: ribosomal protein kinase S6 and 4E-binding protein 1 [44,45]. Insufficient MPS predisposes to loss of muscle mass, with a major impact on type II, fast-twitch fibers [46,47,48]. These fibers are rich in myosin ATPase and glycolytic enzymes with high catalytic activity [46,47,48], allowing them to generate strength rapidly. Hence, it is reasonable to assume that protein intake could be associated with physical performance, and therefore muscle power.

However, this view has not been consistently supported. Isanejad et al. [49] found that a set of physical performance tests, including 5STS, balance, and walking speed, were not associated with protein intake after adjusting the analysis for several covariables. Ten Haaf et al. [36] observed that daily protein intake was not associated with SPPB scores, walking speed, chair rise ability, or isometric handgrip strength. Coelho-Junior et al. [50] found that BW-adjusted daily protein intake was not associated with 5STS performance in Italian older adults. A recent meta-analysis reported that protein intake higher than the RDA was cross-sectionally, but not longitudinally associated with walking speed [31]. These findings are in keeping with our observations, given that mobility, chair rise ability, and balance tests have a high muscle power component [19,51,52,53,54,55].

A potential explanation for our results is based on the pattern of protein distribution. The optimal strategy to provide the required amounts of AAs necessary to properly stimulate MPS is still under debate. Some authors have proposed that a pulse-feeding protein intake through the consumption of a high-protein meal, might saturate the splenic sequestration of AAs, thereby increasing their availability for MPS [36]. However, AA bioavailability seems to reach a plateau at approximately 30 g of protein per meal, after which AAs are oxidated [56,57]. Hence, other researchers argued that a spread feeding pattern with at least 30 g of dietary protein during the main meals could be a more effective strategy to sustain muscle anabolism [36].

In this regard, Farsijani et al. [35] found that older adults with a more spread protein distribution across the main meals were stronger and had better mobility than those with a pulse-feeding intake. Similar findings were reported by Ten Haff et al. [36] in Dutch older adults. In the present study, protein was mostly consumed at lunch, with a smaller percentage ingested at dinner and an even smaller amount at breakfast.

Taken together, these observations suggest that a large and disproportional consumption of proteins at lunch might contribute to whole-body protein oxidation and muscle protein breakdown, consequently favoring muscle atrophy and loss of muscle power. Notably, not all studies support an association between protein distribution and physical function [37,58]. Moreover, a further analysis was conducted to examine the association between protein distribution and power-related parameters (Appendix A), finding no significant associations. Additional studies are necessary to confirm our speculations.

Our sample was prevalently composed of overweight people with a high consumption of protein. Excess protein consumption has long been investigated as a possible risk factor for chronic kidney disease (CKD) [59]. It is suggested that a high intake of protein might enhance glomerular filtration rate, sharply increasing the glomerular pressure and/or causing renal hypertrophy and predisposing to the development of renal damage and CKD [59,60]. In this context, the negative association found between lower-limb muscle power measures and BW-adjusted protein might be the result of long-term impairments of kidney function leading to muscle atrophy and dynapenia [61].

However, longitudinal studies have reported that long-term exposure to HPI has no harmful effects on kidney function in healthy people. Herber-Gast et al. [62] examined more than 3,500 Dutch people and did not observe significant longitudinal associations between protein intake and changes in renal function. Rebholz et al. [63] followed 15,055 North American adults for 21 years. The authors observed that dietary acid load—the balance between acid-inducing and base-inducing foods—was significantly associated with the occurrence of CKD. When the analyses were conducted according to the individual components of the dietary acid load, a higher intake of protein was not associated with CKD. Given that none of the participants of our study reported kidney problems on recruitment, our results are unlikely to be explained by renal damage.

The fact that allometric muscle power was associated with BW-adjusted protein intake when data were analyzed using BMI as a categorical variable suggests that other protein-related aspects might underpin the results of our study. In overweight people, increased ingestion of protein might be associated with the intake of other macronutrients that impact neuromuscular function. For instance, high-fat diets significantly reduce the number of synaptic branches, satellite cell abundance, muscle cross-sectional area, and muscle strength in preclinical models [64,65]. Such a scenario suggests that high-fat/high-protein diets could affect the functional and structural components of the neuromuscular system, causing first impairment of allometric muscle power, which represents acceleration, and then of absolute muscle power. However, no data are available to explore this hypothesis.

Our study is not free of limitations. First, no muscle mass measures were collected, which precluded estimation of specific muscle power. Second, participants were not screened for sarcopenia [11] or frailty [12]. Third, only community-dwelling older adults were examined, and extrapolations to other contexts (e.g., institutionalized older adults) should be made with caution. Fourth, regression analysis was not corrected for covariables that might influence the association between muscle power and protein intake, including physical activity levels, diet quality, and sleep. Fifth, the studied population was relatively small, young, physically active, and with a high prevalence of participants whose protein and BCAA intakes were higher than RDA values. Sixth, the 5STS equation has received some criticism [66], and its predictive value toward health parameters and negative events still needs to be established. Seventh, protein sources might impact the association between physical function and protein-related parameters [41]. Animal-based proteins are thought to provide larger amounts of BCAAs than those of vegetable origin [67], thereby producing greater effects on muscle mass and strength. However, studies found a significant association between plant-based protein and physical function [41], which may be mediated by the amount of plant protein consumed. Hence, future studies should examine the impact of protein sources on muscle power. Finally, the results shown in this work are derived from cross-sectional observations.

## 5. Conclusions

Our findings indicate that absolute and allometric muscle power are negatively associated with BW-adjusted protein intake in community-dwelling older adults.

## Figures and Tables

**Table 1 ijerph-19-14579-t001:** Main characteristics of study participants (*n* = 197).

Variables	Mean, Absolute Number	SD, Percentage	Min	Max
Age, y	68.3	6.8	60	99
Female n (%)	163	83		
Weight, kg	68.9	12.0	40.8	108.0
Height, m	154.7	8.1	139.0	184.0
Educational level				
Primary school (4 years)	36	18.3		
Middle school (8 years)	56	28.4		
Secondary school (11 years)	42	21.3		
Undergraduate (14–15 years)	48	24.4		
Graduate (18–21 years)	15	7.6		
BMI, kg/m^2^	28.7	5.0	15.2	43.1
Underweight, n (%)	17	9		
Normal weight, n (%)	52	26		
Overweight, n (%)	128	65		
5STS, s	14.3	6.5	3.0	70.0
Absolute muscle power, W	154.9	67.0	18.0	648.0
Relative muscle power, W/BMI	0.02	0.008	0.000	0.10
Allometric muscle power, W/m^2^	0.64	0.26	0.09	2.70
Energy intake, kcal	1846.1	545.1	636.6	4490.1
Protein, g	106.1	40.9	30.7	228.2
Protein, g/kg	1.58	0.65	0.4	4.0
Protein at breakfast, g/kg	0.18	0.11	0	0.6
Protein at lunch, g/kg	0.83	0.42	0	2.3
Protein at dinner, g/kg	0.35	0.41	0	1.8
Valine, g/kg	5.3	2.1	1.6	11.4
Isoleucine, g/kg	4.7	2.3	0.2	10.9
Leucine, g/kg	8.0	3.1	2.3	17.3

5STS = sit-to-stand test; BMI = body mass index; SD = standard deviation.

**Table 2 ijerph-19-14579-t002:** Person’s correlation analysis for the association between muscle power and protein-related parameters (n = 197).

Variables	Absolute Muscle Power	Relative Muscle Power	Allometric Muscle Power
Kilocalories	−0.039	−0.049	−0.055
Protein	−0.011	−0.031	−0.028
Protein/kg body weight	−0.241 *	−0.120	−0.233 *
Protein at breakfast	−0.134	−0.111	−0.132
Protein at lunch	−0.210 *	−0.092	−0.201 *
Protein at dinner	−0.044	0.018	−0.047
Valine	0.010	−0.009	−0.008
Isoleucine	0.003	−0.016	0.004
Leucine	0.004	−0.015	−0.014

* *p* < 0.05.

**Table 3 ijerph-19-14579-t003:** Linear regression analysis for the association between muscle power and protein-related parameters (n = 197).

Absolute Muscle Power						
	Unadjusted			Adjusted *		
Variables	Unstandardized Coefficients	*p* Value	95% CI	Unstandardized Coefficients	*p* Value	95% CI
Protein intake/body weight	−23.2	0.005	−39.2, −7.2	−27.8	0.01	−50.0, −5.5
Lunch protein intake	−25.7	0.04	−50.3, −1.0	−12.3	0.37	−39.8, 15.1
**Allometric Muscle Power**						
	**Unadjusted**			**Adjusted ***		
**Variables**	**Unstandardized Coefficients**	** *p* ** **Value**	**95% CI**	**Unstandardized Coefficients**	** *p* ** **Value**	**95% CI**
Protein intake/body weight	−0.08	0.007	−0.151, −0.024	−0.08	0.05	−0.174, 0.01
Lunch protein intake	−0.09	0.06	−0.191, 0.004	−0.03	0.48	−0.14, 0.96

* Adjusted for age, body mass index, sex, and kilocalories; CI = confidence interval.

## Data Availability

Data are available upon request.

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
