# Peer review of "Lower-Limb Muscle Power Is Negatively Associated with Protein Intake in Older Adults: A Cross-Sectional Study"

_ijerph, 2022, doi:10.3390/ijerph192114579_

Round 1

Reviewer 1 Report

Lower-Limb Muscle Power is Negatively Associated with Protein Intake in Older Adults: A Cross-Sectional Study

General observation:

This research is attractive since it addresses a topic of interest to the elderly population, especially at this time when the decade of healthy aging is being promoted.

In this context, nutrition (diet/protein intake) and muscle power are important factors to consider.

To improve the information provided by their study, the authors can consider a few observations.

­­­­­­­­­­­­­­­­­­­­The Abstract in its structured format is easy to understand.

Introduction

At line 40 of page 1, after the references, the paragraph can be expanded to clarify that also in sedentary or frail older adults, disease, functional dependency, and disability tend to negatively impact health outcomes, to emphasize the importance of the topic even further.

As a suggestion, I recommend that the authors provide more details regarding the variables considered in Alcázar et al., original study and which generated the equation to predict Lower-Limb Muscle Power. It is essential to include this information in the research in order to provide the reader with more background information. (Page 2 line 54)

Page 2 line 64 It is recommended to mention that the current recommendation for protein intake in active older adults is greater than 1.0, 1.2, and 1.5 grams per kilogram of body weight per day.

As part of the paragraph: page 2, lines 65-70, it is suggested to include the classic and valuable research of Paddon-Jones et al.  (Paddon-Jones D, Rasmussen BB. Dietary protein recommendations and the prevention of sarcopenia. Curr Opin Clin Nutr Metab Care. 2009 Jan;12(1):86-90. doi: 10.1097/MCO.0b013e32831cef8b.)

I recommend that more information be provided regarding the methodological aspects of Gingrich et al.'s research in order to determine the relationship between protein intake and muscle mass, strength, and power. (Page 2 lines 73-75).

It is also strongly recommended that Lopez et al.'s study be taken into consideration. Losa-Reyna J, Alcazar J, Rodríguez-Gómez I, et al. Low relative mechanical power in older adults: An operational definition and algorithm for its application in the clinical setting. Exp Gerontol. 2020 Dec;142:111141. doi: 10.1016/j.exger.2020.111141.

Material and methods

2.2. Anthropometric measures

Page 2 lines 94-95. Was the WHO classification used to categorize the body mass index (BMI)? It is recommended that you include a bibliographic reference.

Furthermore, the Lipschitz classification may be used to determine the percentage of older adults who are underweight. (DA. Screening for nutritional status in the elderly. Prim Care. 1994;21(1):55-67.)

Results

Page 3 line 137

For a better understanding of the characteristics of the population, it is recommended to add the following information to Table 1: marital status, educational level, and percentage of participants in each BMI category (WHO/ Lipschitz classification).

Is it possible to include information on the source of protein consumed in table 1 (animal or vegetable origin)?

Page 3 line 134: I would like you to review the structure of this paragraph: 35]. On the other hand, mthe ean BW-adjusted protein intake was higher than the RDA

The tables 2 and 3 are clear and without any problems

Discussion

It is highly recommended that the research of Losa-Reyna J et al (mentioned previously) be considered, since it may provide additional information, because there are only a few studies that discuss this interesting topic.

Author Response

Reviewer 1

General observation:

This research is attractive since it addresses a topic of interest to the elderly population, especially at this time when the decade of healthy aging is being promoted. In this context, nutrition (diet/protein intake) and muscle power are important factors to consider. To improve the information provided by their study, the authors can consider a few observations.

 Answer: Thank you for your comment. We are glad that our research was attractive.

Introduction

At line 40 of page 1, after the references, the paragraph can be expanded to clarify that also in sedentary or frail older adults, disease, functional dependency, and disability tend to negatively impact health outcomes, to emphasize the importance of the topic even further.

Answer: Thank you for your suggestion. The paragraph has been expanded (lines: 31-40).

As a suggestion, I recommend that the authors provide more details regarding the variables considered in Alcázar et al., original study and which generated the equation to predict Lower-Limb Muscle Power. It is essential to include this information in the research in order to provide the reader with more background information. (Page 2 line 54)

Answer: More details about the work of Alcázar et al. were provide in lines: 51-57.

Page 2 line 64 It is recommended to mention that the current recommendation for protein intake in active older adults is greater than 1.0, 1.2, and 1.5 grams per kilogram of body weight per day.

Answer: Thank you for your suggestion. We included RDA for physical active older adults (lines 64-67).

As part of the paragraph: page 2, lines 65-70, it is suggested to include the classic and valuable research of Paddon-Jones et al.  (Paddon-Jones D, Rasmussen BB. Dietary protein recommendations and the prevention of sarcopenia. Curr Opin Clin Nutr Metab Care. 2009 Jan;12(1):86-90. doi: 10.1097/MCO.0b013e32831cef8b.)

Answer: Thank you for mentioning this valuable work. It was cited in the text. 

I recommend that more information be provided regarding the methodological aspects of Gingrich et al.'s research in order to determine the relationship between protein intake and muscle mass, strength, and power. (Page 2 lines 73-75).

Answer: We provided more information about the study of Gingrich et al. (lines 78-85).

It is also strongly recommended that Lopez et al.'s study be taken into consideration. Losa-Reyna J, Alcazar J, Rodríguez-Gómez I, et al. Low relative mechanical power in older adults: An operational definition and algorithm for its application in the clinical setting. Exp Gerontol. 2020 Dec;142:111141. doi: 10.1016/j.exger.2020.111141.

Answer: Thank you for your suggestion. The work of Losa-Reyna et al. has been cited in the manuscript.

Material and methods

2.2. Anthropometric measures

Page 2 lines 94-95. Was the WHO classification used to categorize the body mass index (BMI)? It is recommended that you include a bibliographic reference. Furthermore, the Lipschitz classification may be used to determine the percentage of older adults who are underweight. (DA. Screening for nutritional status in the elderly. Prim Care. 1994;21(1):55-67.)

Answer: Dear Reviewer, thank you for your comment. We used the article of Lipschitz to classify BMI in older adults. Participants distribution according to BMI classification might be observed in Table 1 and according to sex in SM1. In addition, we reconducted multilinear regression using BMI as continuous and categorical variables (SM2).

Results

Page 3 line 137

For a better understanding of the characteristics of the population, it is recommended to add the following information to Table 1: marital status, educational level, and percentage of participants in each BMI category (WHO/ Lipschitz classification).

Answer: Educational level and BMI classification were included in Table 1. We do not have data regarding marital status.

Is it possible to include information on the source of protein consumed in table 1 (animal or vegetable origin)?

Answer: Unfortunately, we do not have this data. We included it as a limitation of the present study (lines 282-287).

Page 3 line 134: I would like you to review the structure of this paragraph: 35]. On the other hand, mthe ean BW-adjusted protein intake was higher than the RDA

Answer: We are sorry for this typo. The paragraph was reviewed.

Discussion

It is highly recommended that the research of Losa-Reyna J et al (mentioned previously) be considered, since it may provide additional information, because there are only a few studies that discuss this interesting topic.

Answer: The work of Losa-Reyna has been cited (reference 23).

Reviewer 2 Report

Dear authors, 

I reviewed your article and I have some suggestions:

1. At Abstract remove headings.

2. Mention clearly the aim of the study at the end of Introduction.

3.  Study participants (3.1) must be moved at 2.1 - Participants, including table 1 information from AGE to SEX (also include in the table data about male because now there some data about female). From 5STS to the end of the table will be a distinct table at Results.

4. Offer more details at 3.2 and 3.3 concerning obtained results.

5. Include figure A1 at Discussion where you mentioned in the text (after line 211).

Author Response

Reviewer 2

Dear authors, I reviewed your article and I have some suggestions:

  1. At Abstract remove headings.

Answer: Headings were removed.

  1. Mention clearly the aim of the study at the end of Introduction.

Answer: The last paragraph has been reviewed (lines 86-90).

  1. Study participants (3.1) must be moved at 2.1 - Participants, including table 1 information from AGE to SEX (also include in the table data about male because now there some data about female). From 5STS to the end of the table will be a distinct table at Results.

Answer: Dear Reviewer, we reported our data according to the STROBE statement. It recommends that participants characteristics are described in the Results section. Participants data according to sex are available in SM1. We politely decided to maintain 5STS results in the Table 1.

  1. Offer more details at 3.2 and 3.3 concerning obtained results.

Answer: Sections 3.2 and 3.3 were reviewed.

  1. Include figure A1 at Discussion where you mentioned in the text (after line 211).

Answer: Journal guidelines are mandatory regarding the position of supplementary material.

Reviewer 3 Report

Dear authors,

Your research is valuable in terms of its subject, scope and content. The tests of the research method could have been more extensive. However, I think that these tests may be sufficient for evaluation, since you are doing your research on people over 60. I think your research can be published on ijerp after the minor corrections I have mentioned below.

Introduction

-Please add your hypothesis in the last paragraph under the purpose sentence.

-Please try to create an experimental design above the participants section in the method section. What time are the subjects? at what intervals? What tests have you subjected to? Were there any of the subjects who participated in the research and did not complete the study? (This fix is not mandatory, but it will make your research more valuable if you do).

-Please fix some English grammatical errors.

Kind Regards.

Author Response

Your research is valuable in terms of its subject, scope and content. The tests of the research method could have been more extensive. However, I think that these tests may be sufficient for evaluation, since you are doing your research on people over 60. I think your research can be published on ijerp after the minor corrections I have mentioned below.

Introduction

-Please add your hypothesis in the last paragraph under the purpose sentence.

Answer: The hypothesis of the present study was included in lines 89-90.

-Please try to create an experimental design above the participants section in the method section. What time are the subjects? at what intervals? What tests have you subjected to? Were there any of the subjects who participated in the research and did not complete the study? (This fix is not mandatory, but it will make your research more valuable if you do).

Answer: We include a description of the study design in page, lines. Information about the number of recruited, included and excluded participants are available in lines 92-95.

Kind Regards.

Reviewer 4 Report

This article entitled "Lower-Limb Muscle Power is Negatively Associated with Protein Intake in Older Adults: A Cross-Sectional Study" has an interesting topic. However, this article has serious problems.

Main concerns:

1. As the authors mention the limitations in this article, this study has serious issues with the analyses. There must be several confounders for the linear regression examining the associations between muscle power and protein intake. Those issues can cause misleading readers about understanding the results of this study. The authors should have planned to collect information on the confounders.

2. Related to Comment 1, the authors are not able to explain why dietary protein intake is associated with physical function in both the introduction and discussion sections. In L65-66, although the authors mention the BCAAs, Reference 26 did not examine any BCAAs possibility for the improvements in physical function. The explanation of the mechanism is very important here because this study can not do it by itself.

3. Please properly cite references. For example, in L68-69, the authors say that protein distribution is more important than total protein intake. However, those references did not confirm the superiority of protein distribution to total protein intake. This reviewer doubts some citations in this study. In addition, the authors did not examine whether the protein distribution is associated with outcomes in this study. This can be verbose.

4. The authors seem to use Pearson's correlation to find the reasons for confounding variables. This is not a proper way to conduct multiple regression models. The authors should have confirmed the fittable variable by reading previous studies before conducting this study.

5. To use the data on nutrient intakes, adjusting energy is recommended because nutrient intakes are strongly correlated with energy intake. Otherwise, we can not purely detect the impact of nutrient intakes on outcomes. 

Author Response

This article entitled "Lower-Limb Muscle Power is Negatively Associated with Protein Intake in Older Adults: A Cross-Sectional Study" has an interesting topic. However, this article has serious problems.

Main concerns:

  1. As the authors mention the limitations in this article, this study has serious issues with the analyses. There must be several confounders for the linear regression examining the associations between muscle power and protein intake. Those issues can cause misleading readers about understanding the results of this study. The authors should have planned to collect information on the confounders.

Answer: Dear Reviewer, as you mentioned, several confounders were not collected, and we recognized this aspect as a limitation of the present study (lines 272-289).

  1. Related to Comment 1, the authors are not able to explain why dietary protein intake is associated with physical function in both the introduction and discussion sections.

Answer: The mechanisms underlying the possible association between protein intake and physical function are discussed in lines 202-211.

In L65-66, although the authors mention the BCAAs, Reference 26 did not examine any BCAAs possibility for the improvements in physical function. The explanation of the mechanism is very important here because this study can not do it by itself.

Answer: Please, observe that we did not associate reference 26 with physical function, but with muscle mass. In addition, we mention this reference to support our assumptions about leucine.

  1. Please properly cite references. For example, in L68-69, the authors say that protein distribution is more important than total protein intake. However, those references did not confirm the superiority of protein distribution to total protein intake. This reviewer doubts some citations in this study. In addition, the authors did not examine whether the protein distribution is associated with outcomes in this study. This can be verbose.

Answer: We would kindly ask the Reviewer to be respectful with the comments and assumptions. Please, observe that our references support our assumptions: Farsijani et al. (28515070) found that more-evenly distributed protein intake, independent of the total quantity, was associated with a higher muscle-strength score in both sexes throughout follow-up. It was also associated with a greater mobility score, but only in men and only before adjustment for covariates. Strength and mobility rates of decline were not affected by protein-intake distribution in either sex. Ten Haaf et al. () noted that more spread protein distribution was related to a higher gait speed as opposed to the intermediate distribution group (β = −0.42, p = 0.035, Table 2. However, no significant associations were noted between absolute protein intake and physical function. Gingrich et al. (29240672) mentioned that: In men, a weak correlation between the mean CV and SMI (p = 0.043) was found, indicating an inverse association between evenness of protein intake and muscle mass. Similar to Ten Haff, authors did not observed associations between absolute protein intake and physical function. Protein distribution was examined in Table 1 and SM2.

  1. The authors seem to use Pearson's correlation to find the reasons for confounding variables. This is not a proper way to conduct multiple regression models. The authors should have confirmed the fittable variable by reading previous studies before conducting this study.

Answer: This method was used to test the association between the variables, as a chi-square test. It is useful when the study is not a result of a priori project created to investigate the association between two variables, in this case protein intake and muscle power. We have recognized it as a limitation of the present study.

  1. To use the data on nutrient intakes, adjusting energy is recommended because nutrient intakes are strongly correlated with energy intake. Otherwise, we can not purely detect the impact of nutrient intakes on outcomes. 

Answer: Thank you for your comment. We reanalyzed the data using total kilocalories intake per day. Results are available in Tables 1, 2 and 3, and in SM2.